# Protective Effects of Tea Tree Oil on Inflammatory Injury of Porcine Intestinal Epithelial Cells Induced by Lipopolysaccharide In Vitro

**DOI:** 10.3390/ani14172577

**Published:** 2024-09-04

**Authors:** Li Dong, Qingqing Yuan, Guangzhi Qiu, Yongsheng Zhang, Hongrong Wang, Lihuai Yu

**Affiliations:** 1College of Animal Science and Technology, Yangzhou University, No. 48 of East Wenhui Road, Yangzhou 225009, China; donglijiayou@126.com (L.D.); mz120231569@stu.yzu.edu.cn (Q.Y.); mz120221559@stu.yzu.edu.cn (G.Q.); jackchang2193@139.com (Y.Z.); hrwang@yzu.edu.cn (H.W.); 2Institute of Animal Nutrition, Sichuan Agricultural University, Chengdu 611130, China

**Keywords:** pig, lipopolysaccharide, intestinal epithelial cells, IPI-2I, tea tree oil, inflammation

## Abstract

**Simple Summary:**

Weaning causes a breakdown of intestinal barrier function and thus easily induces diarrhea in pigs. Establishing a proper cell model is helpful for exploring the underlying mechanisms and for rapidly exploiting new feed additives on a large scale. An LPS-induced immune-stressed IPI-2I cell model for piglets was generated and the regulatory effects of TTO on inflammatory injury were studied. The results showed that 50 μg/mL LPS stimulated for 6 h can be used to establish an immune-stressed cell model in IPI-2I cell lines, and 0.03% TTO treatment for 6 h alleviated inflammatory injury in the intestinal epithelial cells of pigs. In addition, TTO might suppress the inflammatory injury to intestinal epithelial cells caused by LPS stimulation through TLR4/NF-κB signaling.

**Abstract:**

Tea tree oil (TTO) improves the intestinal mucosal immunity of weaning piglets, but its underlying mechanism is not clear. We hypothesized that TTO may alleviate inflammatory injury by regulating the function of intestinal epithelial cells. Ileum epithelial cells (IPI-2I) were chosen and an inflammatory injury cell model was generated. The cell viability, cytokine secretion, and gene expression of TLR4 and NF-κB were measured to further evaluate the effects of TTO on the inflammatory injury in immune-stressed cells. The results showed that lipopolysaccharide (LPS; content: ≥30 μg/mL; time: 3 h, 6 h, or 9 h) decreased cell viability (*p* < 0.01), and 50 μg/mL LPS stimulated for 6 h resulted in an increased secretion of proinflammatory cytokines and a dramatically decreased secretion of anti-inflammatory cytokines (*p* < 0.05) in IPI-2I cells. Concentrations of 0–0.05% of TTO improved cell viability, while the 0.03% TTO treatment resulted in the highest cell viability and alleviated LPS-induced cell death (*p* < 0.01). In addition, 0.03% TTO alleviated the LPS-induced increase in the gene expression of IL-1β, TNFα, and IFNγ, as well as the decrease in the expression of IL-10 in IPI-2I cells (*p* < 0.05). LPS also upregulated the gene expression of TLR4 and NF-κB (*p* < 0.05); while TTO supplementation alleviated this effect (*p* < 0.05), 0.03% and 0.05% TTO supplementation had greater effects (*p* < 0.05). In conclusion, 50 μg/mL LPS stimulated for 6 h can be used to establish an immune-stressed cell model in IPI-2I cell lines, and 0.03% TTO treatment for 6 h alleviated inflammatory injury in the intestinal epithelial cells of pigs.

## 1. Introduction

Weaning causes a breakdown of intestinal barrier function and thus easily induces diarrhea in pigs [1]. Establishing a proper cell model to explore the underlying mechanisms and to rapidly exploit new feed additives on a large scale will be helpful for understanding this issue. Both IPEC-J2 and IPI-2I are excellent cell lines for studying the function of the intestinal epithelium of pigs [2,3]. Many studies have used Gram-negative bacteria to establish an immune stress model in IPI-2I cells, but only one study has used LPS from *Salmonella typhimurium* [4]. In that study, the optimal concentration of lipopolysaccharide (LPS) was 1 μg/mL, the stimulation times were 30 min, and 1, 2, 3, and 4 h. The authors reported that cytokine secretion changed but did not determine the best stimulation time [4]. In this study, we established an LPS-induced immune-stressed model in IPI-2I cells on the basis of cell viability, defined as the number of healthy cells in a cell calculation system, as determined using a Cell Counting Kit (CCK-8), along with cytokine secretion and cytokine gene expression.

Tea tree oil (TTO), extracted from *Melaleuca alternifolia*, has broad-spectrum antimicrobial, anticancer, and antioxidant properties [5]. Our previous studies suggested that TTO could improve the intestinal mucosal immunity of weaning piglets [6]. However, its underlying mechanism is not clear. We hypothesized that TTO might improve the intestinal barrier function through alleviating inflammatory injury in the intestinal epithelial cells of pigs. An LPS-induced immune-stressed IPI-2I cell model for piglets was generated and the regulatory effects of TTO on inflammatory injury were studied. These results will help elucidate the underlying mechanisms of intestinal inflammatory injury and facilitate the use of new feed additives.

## 2. Materials and Methods

### 2.1. Cell Line Culture

The IPI-2I cell line derived from the ileum of pigs, was purchased from Jennio Biotechnology Company (No. JN0-02001). The cells were maintained in complete culture medium consisting of Dulbecco’s modified Eagle Medium (DMEM)/Ham’s F12 medium (No. 11965092, Gibco, San Francisco, CA, USA) supplemented with 10% fetal calf serum (FBS, No. 10091, Gibco) and 2% penicillin–streptomycin (No. P1400, Solarbio, Beijing, China). The cell lines were seeded onto 96-well tissue culture plates for cell viability analysis, and onto 24-well or 6-well tissue culture plates with 2 × 10^5^ cells or 2 × 10^6^ cells per well in complete culture medium for the measurement of gene expression, cytokine secretion, and cell apoptosis. The cells were allowed to adhere for 24 h before being fed every two to three days. The cells were sub-cultured after 3 to 4 days. All cells were cultured in an atmosphere of 5% CO_2_ at 37 °C.

### 2.2. Experimental Design

Experiment 1: Lipopolysaccharide (LPS, No. L6143, *Salmonella enterica serovar typhimurium*) was used to establish an immune-stressed IPI-2I cell (porcine ileum epithelial cells) model. LPS (10, 20, 30, 40, 50, 100, and 200 μg/mL) was added to the cell culture system for 3 h, 6 h, 9 h, or 12 h, respectively. Cell viability was measured using a Cell Counting Kit (CCK-8, No.C0038, Beyotime Btotech Inc., Nantong, China), according to the manufacturer’s instructions.

Experiment 2: Based on the results of the Experiment 1, four groups with different concentrations of LPS were established—the CON group (DMEM), LLPS group (DMEM + 10 μg/mL LPS), MLPS group (DMEM + 50 μg/mL LPS), and HLPS group (DMEM + 100 μg/mL LPS)—with stimulation times of 3 h, 6 h, 9 h, and 12 h. The cell supernatants were collected at different time points for all groups to study cytokine secretion. Additional cells were collected at different time points to study the gene expression of cytokines.

Experiment 3: The regulatory effects of TTO on immune-stressed IPI-2I cells were further studied. The composition of TTO was the same as that published in our previous study [6]. Tea tree oil is a plant essential oil that is difficult to dissolve in high-sugar DMEM. The tea tree oil was filtered through a 0.22 μm bactericidal filter into a 50 mL centrifuge tube and was stored at 4 °C for later use. A DMSO solution with a concentration fraction of 0.01–0.1% TTO was used to stimulate the IPI-2I cells for 3 h, 6 h, 9 h, and 12 h. Cell viability was tested.

Experiment 4: According to the results of the cell viability assay, 0.01%, 0.03%, and 0.05% TTO were used to stimulate the cells for 6 h. Five groups were established—the CON group (DMEM), LPS group (DMEM + 50 μg/mL LPS), LTTO group (DMEM + 50 μg/mL LPS + 0.01% TTO), MTTO group (DMEM + 50 μg/mL LPS + 0.03% TTO), and the HTTO group (DMEM + 50 μg/mL LPS + 0.05% TTO)—with the simultaneous addition of LPS or TTO to the serum-free DMEM, with a treatment time of 6 h. The doses of LPS (50 mg/mL) and the 6 h incubation time were selected based on the results of Experiments 1 and 2, which can be found in the results and discussion section of the manuscript. Cell viability was measured. The cell supernatant and cells were collected for the study of cytokine secretion, cell apoptosis, and gene expression of both cytokines and TLR4/NF-κB signaling.

### 2.3. Cell Viability

IPI-2I cells (six repetitions per treatment) were seeded in a 96-well plate (2 × 10^4^ cells per well), and were placed in 5% CO_2_ at 37 °C for 24 h. The culture medium was then removed; then, the media (fetal calf serum-free) was supplemented with different concentrations of LPS or tea tree oil and was cultured for different time periods. After that, the Cell Counting Kit-8 reagent (CCK-8; Beyotime Biotechnology, Nantong, China) was added and the cells were cultivated for 1 h. Finally, the absorbance values were detected with a microplate reader (wavelength: 450 nm).

### 2.4. Cytokine Secretion

The levels of interleukin (IL)-1β (No. H002), IL-10 (No. H009), tumor necrosis factor-α (TNFα) (No. H052), and interferon-γ (IFN-γ) (No. H025) in the cell supernatant were determined using the enzyme-linked immunosorbent assay (ELISA) (pig-specific), according to the manufacturer’s instructions (Nanjing JianCheng Bioengineering Institute, Nanjing, China). The methods used were the same as in our previous study [6]. The samples were assayed in triplicate. The ELISA results were detected using a microplate ELISA reader at 450 nm and were expressed as nanograms per liter of supernatant.

### 2.5. Cell Apoptosis

A total of 2 × 10^6^ cells were seeded in 6-well tissue culture plates and different concentrations of LPS or TTO were added to the fetal/calf serum-free medium. After 6 h of stimulation, the cells were collected in 15 mL centrifuge tubes and were washed twice with PBS. In accordance with the procedure of the cell apoptosis kit, 10× binding buffer was diluted to 1× binding buffer using PBS, and 500 μL 1× binding buffer was added to the cells. After blending, 100 μL of cells were added to the flow cytometric tube, and the Annexin V-TITC/PI mixture and 400 μL 1× binding buffer were added. The cells were then incubated in the dark for 30 min at room temperature. Finally, cell apoptosis was detected using flow cytometry. The percentage of apoptotic cells was determined from the flow cytometry results. Three repeats were conducted for each group. The results were analyzed by one-way ANOVA.

### 2.6. Gene Expression

The IL-10, TNFα, IFN-γ, TLR4, and NF-κB levels in the cells were measured. The methods for the isolation of RNA and real-time polymerase chain reaction (RT-PCR) were the same as those in our previous study [6]. GAPDH was chosen as a housekeeping gene (β-actin was also used as a control gene to normalize the expression of target genes). The gene-specific primers used are listed in Table 1 and were synthesized by Invitrogen Biotech Co. Ltd. (Shanghai, China). We observed the amplification curve and dissolution curve when we conducted the RT-PCR experiments. The amplification curve was normal, and the dissolution curve was a single peak, which indicated that the primers had good specificity and efficacy. In addition, agar gel electrophoresis was conducted after ordinary gradient PCR. The amplification products of the primers were single and consistent with the expected size, which also indicated that the primer has good specificity. The relative gene expression levels were determined using Step-One software version 2.3 (Applied Biosystems) and were calculated with the 2^−ΔΔCT^ method [7].

### 2.7. Statistical Analysis

Statistical Package for the Social Sciences (SPSS) 19.0 software was used for data analysis. Independent *t* tests were conducted for data analysis of cell viability, and one-way ANOVA was used for the analysis of the other data. Homogeneity analysis of variance was conducted, then an independent *t* test or Duncan analysis was used to compare the differences between the two groups. All the graphs were drawn using Graphpad Prism 7.0. The *p* value indicates the difference among the groups; *p* < 0.05 indicates a significant difference, while *p* < 0.01 indicates the difference is extremely significant.

## 3. Results

### 3.1. Cell Viability after LPS Stimulation

Figure 1 shows the effects of LPS stimulation on the cell viability of IPI-2I cells. Compared with the 0 μg/mL group, the viability of the cells in the ≥30 μg/mL group was significantly lower (*p* < 0.01) after 3 h, 6 h, and 9 h of stimulation with LPS, while the viability of the cells in the 0–50 μg/mL LPS and 0 μg/mL groups was not significantly different (*p* > 0.05) after 12 h of stimulation. In addition, 100 μg/mL and 200 μg/mL LPS significantly reduced the viability of the IPI-2I cells (*p* < 0.01) after 12 h of stimulation.

### 3.2. Cytokine Secretion after LPS Stimulation

Figure 2 illustrates the effects of LPS stimulation on the cytokine secretion in IPI-2I cells. The secretion of IL-1β and IFN-γ in the LPS treatment groups was greater than that in the CON group after 3 h, 6 h, 9 h, or 12 h of stimulation with LPS (*p* < 0.05). When the stimulation time was 3, 9, or 12 h, the IL-1β secretion in the MLPS and HLPS groups was significantly greater than that in the LLPS group (*p* < 0.05); however, when the stimulation time was 6 h, the IL-1β secretion in the MLPS group and the HLPS group were 7.83 times and 9.14 times greater than that in the CON group, which is a much more rapid change than in the groups stimulated with LPS for 3, 9, or 12 h. When the stimulation time was 3, 6, or 12 h, IFN-γ secretion was significantly greater in the MLPS and HLPS groups compared with the LLPS group (*p* < 0.05); however, when the stimulation time was 6 or 9 h, the IFN-γ secretion in the MLPS group were 10.61 and 15.40 times greater than that in the CON group, which is a much more rapid change than in the groups stimulated with LPS for 3 or 12 h. When the stimulation time was 3 or 6 h, the secretion of TNF-α in the LPS treatment group was significantly greater than that in the CON group (*p* < 0.01). When the stimulation time was 3 h, the content of TNF-α in the MLPS group was significantly greater than that in the LLPS and HLPS groups (*p* < 0.01). When the stimulation time was 6 h, the secretion of TNF-α was significantly higher than in the LLPS group (*p* < 0.05). When the stimulation time was 6 h, the IL-10 secretion in the MLPS group was significantly greater than that in the other groups (*p* < 0.05).

### 3.3. Gene Expression of Cytokines after LPS Stimulation

LPS stimulation upregulated (*p* < 0.01) the gene expression of IL-1β, TNF-α, and IFN-γ, and downregulated the gene expression of IL-10 (*p* < 0.01). Compared with those in the other stimulation times, the gene expression levels of IL-1β, TNF-α, and IFN-γ in the IPI-2I cells in the LPS group were higher when the stimulation time was 6 h; the gene expression levels of IL-1β and TNF-α in the MLPS group were even higher than those in the HLPS group (*p* < 0.05) (Figure 3).

### 3.4. Viability of IPI-2I Cells after Tea Tree Oil Treatment

TTO at a concentration of 0–0.05% had a positive effect on the viability of IPI-2I cells. At culture times of 3 h, 9 h, and 12 h, 0.04% TTO significantly increased (*p* < 0.05) the viability of IPI-2I cells. When the TTO concentration was greater than 0.07%, the viability of IPI-2I cells was significantly reduced (*p* < 0.05). Therefore, TTO has a low toxicity concentration range of 0–0.05% for the viability of IPI-2I cells (Figure 4).

### 3.5. Cell Apoptosis

The effects of TTO on cell apoptosis in immune-stressed IPI-2I cells are shown in Figure 5. LPS treatment decreased the viability and increased the early apoptosis (*p* < 0.05) and the late apoptosis (*p* < 0.01) of IPI-2I cells, while TTO alleviated this LPS-induced cell death (*p* < 0.05), and 0.03% TTO treatment had better effects (*p* < 0.05).

### 3.6. Cytokine Secretion and Gene Expression

Figure 6 shows the effects of tea tree oil on the cytokine secretion and gene expression of cytokines in the immune-stressed IPI-2I cells. Compared with those in the CON group, the secretion of IL-1β, TNF-α, and IFN-γ in the LPS group was significantly greater; however, TTO supplementation moderated this increase (*p* < 0.01), and MTTO had greater effects (*p* < 0.05) (Figure 6A). LPS stimulation significantly increased the gene expression of IL-1β, TNF-α, and IFN-γ (*p* < 0.05) but decreased the gene expression of IL10 in IPI-2I cells (*p* < 0.05), while TTO supplementation alleviated these effects (*p* < 0.05) (Figure 6B).

### 3.7. Gene Expression of NF-κB and TLR4

Figure 7 shows the effects of tea tree oil on the gene expression of TLR4 and NF-κB in the immune-stressed IPI-2I cells. LPS stimulation significantly increased the gene expression of TLR4 and NF-κB (*p* < 0.05), while TTO supplementation alleviated this effect (*p* < 0.05) and MTTO and HTTO treatments had greater effects (*p* < 0.05).

## 4. Discussion

TTO improved the intestinal mucosal immunity of weaning piglets [6]. However, the underlying mechanism is not clear. We hypothesized that TTO might improve intestinal barrier function through alleviating inflammatory injury in the intestinal epithelial cells of pigs. Porcine ileum epithelial cells (IPI-2I) were chosen for this study, and LPS was used to establish the immune-stressed cell model. The cell viability, cytokine secretion, cell apoptosis, and gene expression of TLR4 and NF-κB were further measured to study the effects of TTO on inflammatory injury in immune-stressed epithelial cells in piglets. The results of this study will help to elucidate the underlying mechanisms by which TTO regulates intestinal health. The immune-stressed IPI-2I cell model established here will aid in the selection of proper regulatory nutrients for improving intestinal barrier function in both piglets and human infants.

Cell viability reflects cell growth. In this study, LPS stimulation (content: ≥30 μg/mL; time: 3 h, 6 h or 9 h) decreased the viability of IPI-2I cells. When the stimulation time was 12 h, LPS at concentrations of 100 μg/mL and 200 μg/mL significantly reduced the viability of IPI-2I cells. Previous studies have demonstrated that LPS stimulation (1.5 μg/mL or 100 μg/mL; 24 h) decreased the viability of porcine IPEC-J2 cells [8,9]. In this study, stimulation with 50 μg/mL LPS for 6 h improved the secretion of IL-1β, TNF-α, and IFN-γ, and decreased the secretion of IL-10, so as to alter the gene expression of these cytokines. A previous study revealed an increased gene expression of TNF-α after 3 h, and an upregulated gene expression of IL-1β after 2 h, in IPI-2I cells [4]. Similar results have been demonstrated in the studies of IPEC-J2 cells. LPS stimulation at a concentration of 1.5 μg/mL for 24 h significantly increased the secretion of the proinflammatory cytokines IL-1β and TNF-α in porcine IPEC-J2 cells [8]. LPS stimulation at a concentration of 10 μg/mL for 6 h also increased TNF-α secretion in IPEC-J2 cells [10]. IFN-γ is an immune interferon that can inhibit the proliferation of viruses in pigs [11]. Hence, 50 μg/mL LPS stimulation for 6 h is suitable for establishing an immune-stressed model in IPI-2I cells.

TTO, extracted from *Melaleuca alternifolia*, has broad-spectrum antimicrobial, anticancer, and antioxidant properties [12]. Our previous study suggested that 0.01% TTO supplementation could improve the intestinal barrier function of weaning piglets [6]. In this study, the underlying mechanism was further studied using an immune-stressed intestinal epithelial cell model. The results of this study showed that TTO at a concentration of 0–0.05% had a positive effect on the viability of IPI-2I cells. Our results are similar with a previous study. The cell viability rate of IPEC-J2 cells significantly increased when the concentration of TER was between 0.002% and 0.016%, that is, when the concentration of TTO was between 0.006% and 0.048% [13]. Considering the other factors including the effects of TTO on the LPS-induced cell apoptosis and cytokine secretion in IPI-2I cells (discussed in the following two paragraphs), 0.03% TTO supplementation is recommended for immune-stressed cell models. On the other hand, 0.01% TTO supplementation was recommended for weaning piglets in our previous study [6]. Whether one-third of a dose of the nutrient supplement in cell models could be used as a dose of the nutrient supplement in animal production requires further research. If this is proven to be correct, it will accelerate the selection of new additives to be employed in piglet production.

The effects of TTO on cell death are closely associated with its dose. A previous study suggested that TTO and TER could inhibit the proliferative activity of tumor cells at a concentration of 0.2% [14]. Another study demonstrated that TTO had little toxicity to epithelial cells at concentrations lower than 100 μg/mL, but at concentrations higher than 100 μg/mL, TTO rapidly decreased cell viability [15]. TER, the main functional constituent of TTO, can also alleviate LPS-induced damage in IPEC-J2 cells [13]. However, the effects of TTO on the death of intestinal epithelial cells (especially IPI-2I cells) in piglets have rarely been reported. The results of our study suggested that 0.03% TTO treatment had the greatest effect on alleviating the LPS-induced cell death in IPI-2I cells. Increased intestinal epithelial cell death may disrupt the mucosal immune homeostasis and increase inflammation [16]. The results of our study suggested that an appropriate dose of TTO could help re-establish the cell homeostasis affected by LPS stimulation in intestinal epithelial cells.

LPS is a potent stimulator of the production of interleukins, including IL-1, IL-6, and TNF-α [17]. A previous study showed that after LPS stimulation, the gene expression of both IL-1β and TNF-α were upregulated in the IPI-2I and IPEC-J2 cell lines [4]. Our results agree with previous studies, which showed that LPS caused the upregulation of proinflammatory cytokines (IL-1β, TNF-α, and IFN-γ). Tea tree oil can inhibit the production of proinflammatory cytokines in LPS-stimulated human macrophages [18]. TTO has been reported to inhibit the secretion of proinflammatory cytokines in weaning-stressed piglets [6,19]. The anti-inflammatory effect of TTO might be mediated by the hypothalamic–pituitary–adrenal axis [20]. In this study, 0.03% TTO alleviated the upregulated secretion of proinflammatory cytokines and downregulated the secretion of anti-inflammatory (IL10) cytokines in immune-stressed IPI-2I cells. The effects of TTO on the gene expression of cytokines may be due to the action of terpenes (the main functional component of TTO) on the transcription factor NF-κB [21], and we further discuss the gene expression of NF-κB in the following paragraph. These results suggested that TTO might improve the intestinal barrier function of weaning piglets through the regulation of the cytokine secretion of intestinal epithelial cells.

LPS can be recognized by TLR4 and then initiate NF-κB signaling, which may cause an inflammatory response and organ injury [22]. A previous study demonstrated that LPS stimulation could activate the expression of TLR4 and NF-κB in intestinal epithelial cell lines in humans [23]. In this study, LPS stimulation significantly increased the gene expression of TLR4 and NF-κB, while TTO supplementation alleviated this effect. TTO can also decrease the *E. coli* LPS-induced activation of NF-κB in human macrophages [18]. Our results suggested that TTO might suppress the inflammatory injury to intestinal epithelial cells caused by LPS stimulation through TLR4/NF-κB signaling.

In conclusion, stimulation with 50 μg/mL LPS for 6 h is suitable for establishing an immune-stressed IPI-2I cell model, and treatment with 0.03% TTO for 6 h alleviates the inflammatory injury caused by LPS in the intestinal epithelial cells of pigs.

## Figures and Tables

**Figure 1 animals-14-02577-f001:**
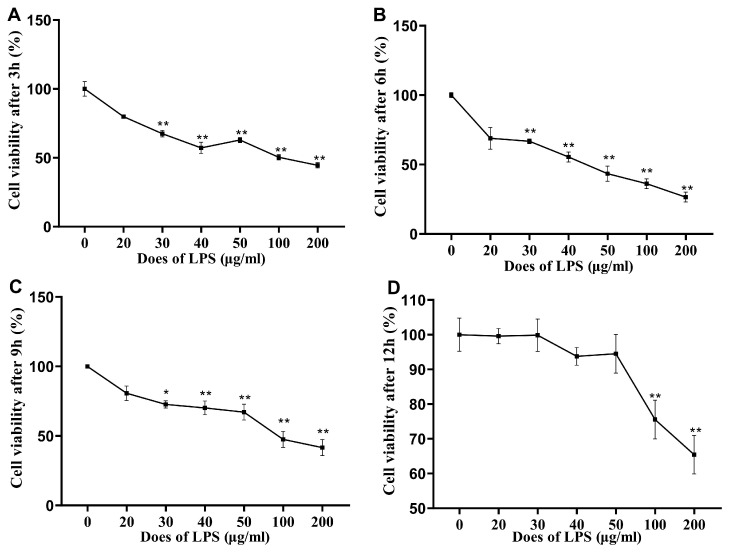
Effects of LPS stimulation on the cell viability of IPI-2I cells. The cell viability of IPI-2I cells stimulated by 0, 20, 30, 40, 50, 100 and 100 μg/mL LPS for 3 h is shown in (**A**), the cell viability of IPI-2I cells stimulated by different doses of LPS for 6 h, 9 h and 12 h are shown in (**B**–**D**), respectively. Data are expressed as mean ± SEM, n = 6. All data were analyzed by independent *t* test. *p* value indicates the difference among the groups; *p* < 0.01 indicates an extremely significant difference, while *p* < 0.05 indicates a significant difference. Groups with letters of * had significant differences compared to the group without LPS stimulation; * *p* < 0.05, ** *p* < 0.01.

**Figure 2 animals-14-02577-f002:**
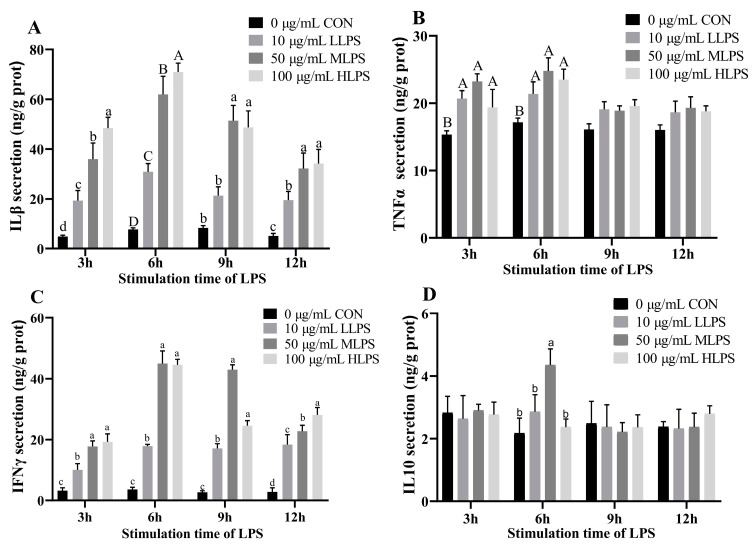
Effects of LPS stimulation on cytokine secretion in IPI-2I cells. The content of IL-1β secreted by cells after LPS stimulation for 3 h, 6 h, 9 h, 12 h is shown in (**A**). The content of TNFα, IFNγ, and IL-10 secreted by cells after LPS stimulation for 3 h, 6 h, 9 h, 12 h are shown in (**B**–**D**), respectively. Data are expressed as mean ± SEM, n = 6. All data were analyzed by one-way ANOVA. *p* value indicates the difference among the groups; *p* < 0.01 indicates an extremely significant difference, while *p* < 0.05 indicates a significant difference. Groups with different letters had significant differences; different lowercase letters (a, b, c, d.) mean *p* < 0.05; different uppercase letters (A, B, C, D) mean *p* < 0.01.

**Figure 3 animals-14-02577-f003:**
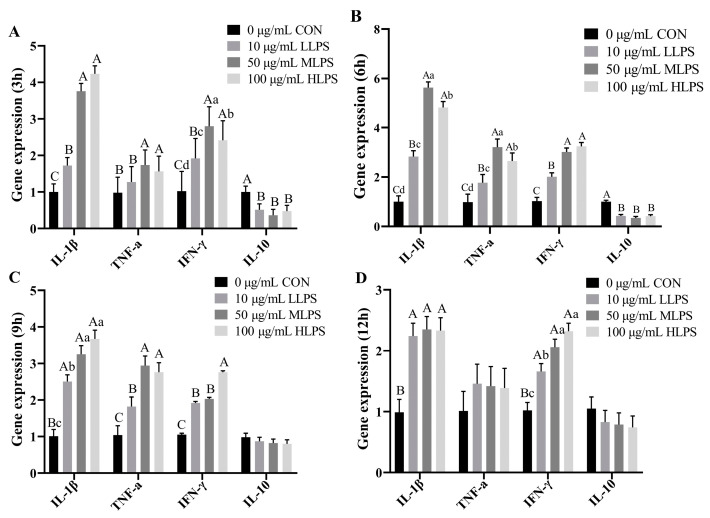
Effects of LPS stimulation on the gene expression of cytokines in IPI-2I cells. The gene expression of IL-1β, TNFα, IFNγ, and IL-10 of the cells after LPS stimulation for 3 h is shown in (**A**). The gene expression of the cytokines in the cells after LPS stimulation for 6 h, 9 h, 12 h are shown in (**B**–**D**), respectively. Data are expressed as mean ± SEM, n = 6. All data were analyzed by one-way ANOVA. *p* value indicates the difference among the groups; *p* < 0.01 indicates an extremely significant difference, while *p* < 0.05 indicates a significant difference. Groups with different letters had significant differences; different lowercase letters (a, b, c, d.) mean *p* < 0.05; different uppercase letters (A, B, C) mean *p* < 0.01.

**Figure 4 animals-14-02577-f004:**
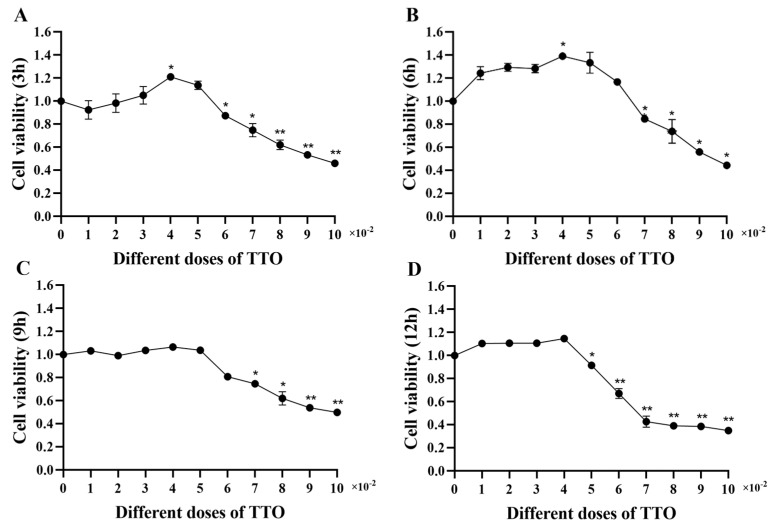
Effects of tea tree oil on the cell viability of IPI-2I cells. Data are expressed as mean ± SEM, n = 6. The cell viability of IPI-2I cells stimulated by 0.01–0.1% TTO for 3 h is shown in (**A**). The cell viability of IPI-2I cells stimulated by different doses of TTO for 6 h, 9 h and 12 h are shown in (**B**–**D**), respectively. All data were analyzed by independent *t* test. *p* value indicates the difference among the groups; *p* < 0.01 indicates an extremely significant difference, while *p* < 0.05 indicates a significant difference. Groups with letters of * had significant differences compared to the group without LPS stimulation; * *p* < 0.05, ** *p* < 0.01.

**Figure 5 animals-14-02577-f005:**
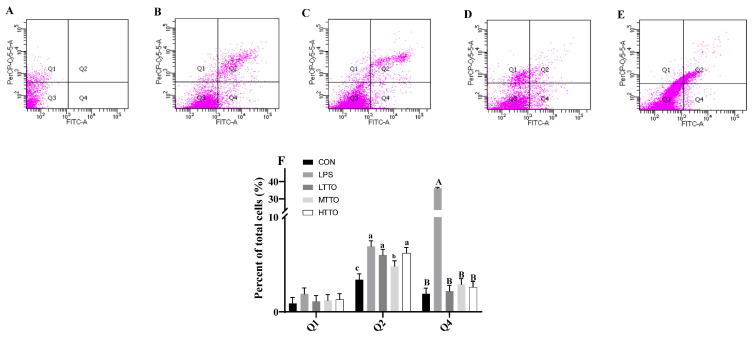
Effects of tea tree oil on cell apoptosis in immune-stressed IPI-2I cells. (**A**) shows the results of flow cytometry for the control group, (**B**) shows the results for the LPS group, (**C**) shows the results for the LPS + LTTO group, (**D**) shows the results for the LPS + MTTO group, and (**E**) shows the results for the LPS + HTTO group. Q1 is AnnexinV-TITC^−^/PI^+^, which represents the number of necrotic cells; Q2 is AnnexinV-TITC^+^/PI^+^, which represents the number of late apoptotic cells; Q3 is AnnexinV-TITC^−^/PI^−^, which represents the live cell quantity; Q4 is AnnexinV-TITC^+^/PI^−^, which represents the number of early apoptotic cells. (**F**) shows the percent of total cells in Q1, Q2, and Q4. Cell percent in different regions was analyzed by one-way ANOVA. *p* value indicates the difference among the groups; *p* < 0.01 indicates an extremely significant difference, while *p* < 0.05 indicates a significant difference. Groups with different letters had significant differences; different lowercase letters (a, b, c) mean *p* < 0.05; different uppercase letters (A, B) mean *p* < 0.01.

**Figure 6 animals-14-02577-f006:**
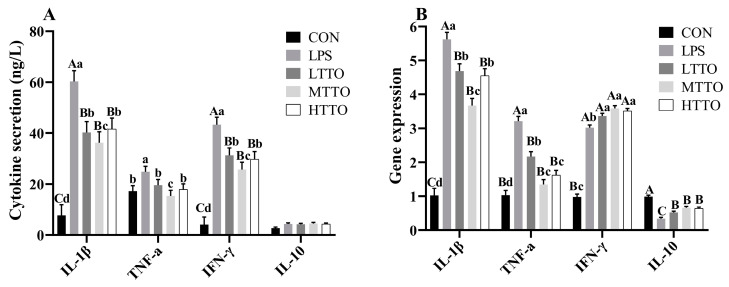
Effects of tea tree oil on the secretion and gene expression of cytokines in immune-stressed IPI-2I cells. The effects of TTO on the secretion of IL-1β, TNFα, IFNγ, and IL-10 in LPS stimulated cells is shown in (**A**). The gene expression of IL-1β, TNFα, IFNγ, and IL-10 in LPS stimulated cells is shown in (**B**). Data were analyzed by one-way ANOVA. *p* value indicates the difference among the groups; *p* < 0.01 indicates an extremely significant difference, while *p* < 0.05 indicates a significant difference. Groups with different letters had significant differences; different lowercase letters (a, b, c, d) mean *p* < 0.05; different uppercase letters (A, B, C) mean *p* < 0.01.

**Figure 7 animals-14-02577-f007:**
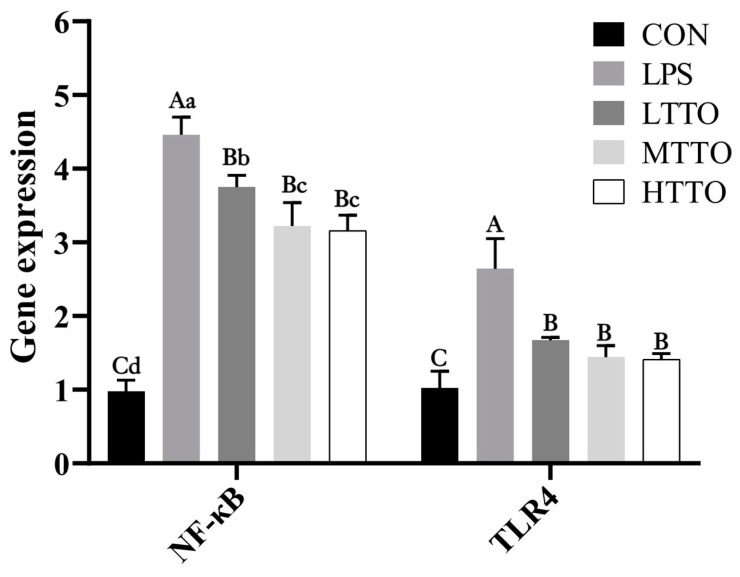
Effects of tea tree oil on the gene expression of TLR4 and NF-κB in the immune-stressed IPI-2I cells. Data were analyzed by one-way ANOVA. *p* value indicates the difference among the groups; *p* < 0.01 indicates an extremely significant difference, while *p* < 0.05 indicates a significant difference. Groups with different letters had significant differences; different lowercase letters (a, b, c, d) mean *p* < 0.05; different uppercase letters (A, B, C) mean *p* < 0.01.

**Table 1 animals-14-02577-t001:** Primer sequences of the tested genes.

Gene	Primer Sequence (5′-3′)	Fragment, bp
GAPDH	5′ GTCGGAGTGAACGGATTTGGC 3′ 5′ GGAGGTCAATGAAGGGGTCA 3′	106
IL-1β	5′ GTGGCAGGACCTACACTCTTC 3′ 5′ TTCCTTCAGAATGCCGTCCTC 3′	115
IL10	5′ GTGGCAGCCAGCATTAAGTC 3′ 5′ AACTCTTCACTGGGCCGAAG 3′	103
IFN-γ	5′ GGCCATTCAAAGGAGCATGGA 3′ 5′ TCACTGATGGCTTTGCGCT 3′	144
TNF-α	5′ GCCCTTCCACCAACGTTTTC 3′ 5′ CAAGGGCTCTTGATGGCAGA 3′	97
NF-κB p65	5′ AGATCTTCCTGCTGTGCGAC 3′ 5′ GTCGGCTTGTGAAAAGGAGC 3′	98
TLR4	5′ GTGGCCTCCAAGGAACAAGA 3′ 5′ CTGGTGTTCACACGCACAAG 3′	96

## Data Availability

The original contributions presented in the study are included in the article, further inquiries can be directed to the corresponding author.

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
