# Peer review of "Protective Effects of Tea Tree Oil on Inflammatory Injury of Porcine Intestinal Epithelial Cells Induced by Lipopolysaccharide In Vitro"

_animals, 2024, doi:10.3390/ani14172577_

Round 1
Reviewer 1 Report
Comments and Suggestions for Authors
The current study evaluates the effects of LPS and tea tree oil on various measures of immune function and cell viability in vitro in a particular line of porcine epithelial cells. The authors aim to understand better the mechanisms through which TTO can improve intestinal immune function. While there are no major issues with the study design or conduct, the authors failed to present their results logically and scientifically. The manuscript is filled with grammatical and editorial errors, making it difficult to follow the authors' reasoning. Additionally, the discussion section is incomplete and vague, leaving the conclusions of the study unclear.
I strongly recommend that the authors undertake significant revisions to their manuscript, including consulting with an English language expert. They might also consider using AI tools to improve their writing, provided it is allowed by the journal. In its current form, the manuscript is not ready for publication.
Please see my detailed comments attached.

Author Response
Dear Editor,
Thank you very much for your kind letter and comments from the reviewers about our paper entitled "Protective effects of tea tree oil on inflammatory injury of porcine intestinal epithelial cells induced by lipopolysaccharide" (Manuscript ID: animals-3098306), which was submitted to Animals.
We have carefully considered the comments of reviewers and have revised our manuscript according to the advice that we were given.
If you have any more questions about this paper, please contact us without hesitation. Thank you very much and all the best to you.
Sincerely yours,
Li Dong
Response to Reviewer1
General Comments –
The current study evaluates the effects of LPS and tea tree oil on various measures of immune function and cell viability in vitro in a particular line of porcine epithelial cells. The authors aim to understand better the mechanisms through which TTO can improve intestinal immune function. While there are no major issues with the study design or conduct, the authors failed to present their results logically and scientifically. The manuscript is filled with grammatical and editorial errors, making it difficult to follow the authors' reasoning. Additionally, the discussion section is incomplete and vague, leaving the conclusions of the study unclear.
I strongly recommend that the authors undertake significant revisions to their manuscript, including consulting with an English language expert. They might also consider using AI tools to improve their writing, provided it is allowed by the journal. In its current form, the manuscript is not ready for publication.
Please see my detailed comments attached.
Response – Thank you for the suggestion. We have made major revision according to your suggestion to increase the readability of the manuscript. We have added details to the Materials and Methods. In addition, the manuscript has been edited for proper English language, grammar, punctuation, spelling, and overall style by one or more of the highly qualified native English speaking editors at American Journal Experts. The verification code is 37C6-D59E-A9D0-DEEE-5B44.
Comments – 1. Abstract:You need a statement briefly describing your methodology. For example you can say "a gene expression study was conducted to evaluate....."
Response – Thank you for the suggestion, and we have added the describe of the methodology in the abstract. “Ileum epithelial cells (IPI-2I) were chosen and an inflammatory injury cell model was generated. The cell viability, cytokine secretion and gene expression of TLR4 and NF-κB were measured to further evaluate the effects of TTO on the inflammatory injury in immune-stressed cells.”
Comments – 2. inner Change to "underlying".
Response – Thank you for the suggestion, and we have changed “inner” to “underlying”.
Comments – 3. “The results showed that LPS (content: ≥30μg/ml; 16 0.01), time: 3h, 6h or 9h ) decreased the cell viability (P< 50μg/ml LPS stimulated for 6h 17 exhibited sharp secretion of pro- inflammatory cytokines and dramatic decreased secreation of anti-inflammatory cytokines (P<0.05) in IPI-2I cells.”Awkward sentence. It needs revision.
Response – Thank you for the suggestion. We have made revision. The revised sentence is as follow: “The results showed that lipopolysaccharide (LPS; content: ≥ 30 μg/ml; time: 3 h, 6 h or 9 h) decreased cell viability (P<0.01), and 50 μg/ml LPS stimulated for 6 h resulted in increased secretion of proinflammatory cytokines and dramatically decreased secretion of anti-inflammatory cytokines (P<0.05) in IPI-2I cells.”
Comments – 4. “0-0.05% concentration of TTO 19 improved the cell viability, while 0.03% TTO treatment s howed highest cell viability and 20 alleviated LPS caused cell death (P<0.01).”Awkward Statement. Please revise. Do not start your sentence with a number.
Response – Thank you for the suggestion. We have made revision. The revised sentence is as follow: “Concentration of 0-0.05% TTO improved cell viability, while the 0.03% TTO treatment resulted in the highest cell viability and alleviated LPS-induced cell death (P<0.01).”
Comments – 5. up-regulation genes?
Response – Yes, we have made revision to improve the integrity of the sentence. “ In addition, 0.03% TTO alleviated the LPS-induced increase in the gene expression of IL-1β, TNFα, and IFNγ and the decrease in the expression of IL-10 in IPI-2I cells (P<0.05).”
Comments – 6. ILβ, Is this IL-1Beta?
Response – Thank you for the suggestion. We made a mistake here. Yes, this word should be “IL-1β”. And we have made revision in the manuscript.
Comments – 7. “LPS also up-regulated gene expression of TLR4 and NF-κB (P<0.05); while TTO supplementation alleviated this situation (P<0.05), 0.03% and 0.05% TTO supplementation are with better effects (P<0.05).” Awkward statement. Please revise.
Response – Thank you for the suggestion. We have made revision. The revised sentence is as follow: “ LPS also upregulated the gene expression of TLR4 and NF-κB (P<0.05); while TTO supplementation alleviated this effect (P<0.05), 0.03% and 0.05% TTO supplementation had greater effects (P<0.05).”
Comments – 8. “In conclusion, 50 μg/ml LPS stimulated for 6h can be used to 25 build the immune-stress ed cell model in IPI-2I cell lines, and 0.03% TTO treated for 6h 26 alleviated the inflamma tory injury in intestinal epithelial cells of pigs.” badly written. revise.
Response – Thank you for the suggestion. We have made revision. The revised sentence is as follow: “In conclusion, 50 μg/ml LPS stimulated for 6 h can be used to establish an immune-stressed cell model in IPI-2I cell lines and 0.03% TTO treatment for 6 h alleviated inflammatory injury in the intestinal epithelial cells of pigs.”
Comments – 9. “Establishing a proper cell model to explore the inner mechanisms and to rapidly explo it new feed additives on a large scale is the key to solve the problem.” I don't think this statement is scientifically correct. In vitro methods will help you better understand the issue, not solve it.
Response – Thank you for the suggestion, and we have made revision according to your suggestion. “Establishing a proper cell model to explore the underlying mechanisms and to rapidly exploit new feed additives on a large scale will be helpful for understanding this issue.”
Comments – 10. inner——underlying mechanism.
Response – Thank you for the suggestion, and we have changed “inner” to “underlying”.
Comments – 11. “Many researches used gram-negative bacterium to establish the immune stress mod el on IPI-2I cells, but only one research was found using LPS from salmonella typhimurium [4].” awkward statement.
Response – Thank you for the suggestion. We have made revision. The revised sentence is as follow: “Many studies have used gram-negative bacteria to establish an immune stress model in IPI-2I cells, but only one has used LPS from Salmonella typhimurium [4].”
Comments – 12. “viability”,You need to define viability. What do you mean by "viability?
Response – Thank you for the suggestion. We have made revision by defining viability. The revised sentence is as follow: “In this study, we established an LPS-induced immune-stressed model in IPI-2I cells on the basis of cell viability, defined as the number of healthy cells in a cell calculation system, as determined using a Cell Counting Kit (CCK-8), along with cytokine secretion and cytokine gene expression.”
Comments – 13. “Hence, ileum epithelial cells (IPI-2I) were chosen, and lipopolysaccharide (LPS) were us ed to build the immune-stressed cell model, and the cell viability, cytokines secretion, cell apoptosis and gene expression of TLR4 and NF-κB were further measured to study the effects of TTO on inflammatory injury in immune-stressed epithelial cells in piglets. The results of this study will help in the usage of TTO in pig industry. And the established immune-stressed IPI-2I cell model will also help for the nutritional regulation of the intestinal barrier function in both pigs and human in-” This section sounds like the justification for your methodology. This must be in your discussion section, not the introduction.
Response – Thank you for the suggestion. The revision has been made. We have written this part to make it more terse. The revised sentence is as bellow: “A LPS-induced immune-stressed IPI-2I cell model for piglets was generated and the regulatory effects of TTO on inflammatory injury were studied. These results will help elucidate the underlying mechanisms of intestinal inflammatory injury and facilitate the use of new feed additives.”
Comments –14. “The cell lines were seeded onto 96-well tissue culture plates for the cell viability analysis and were seeded onto 24-well or 6-well tissue culture plates with 2×105 cells or 2 ×106 cells per well in complete culture solution” revise.
Response – Thank you for the suggestion. We have made revision. “ The cell lines were seeded onto 96-well tissue culture plates for cell viability analysis and onto 24-well or 6-well tissue culture plates with 2 × 105 cells or 2 × 106 cells per well in complete culture medium for the measurement of gene expression, cytokine secretion and cell apoptosis.”
Comments – 15. “Four groups with different concentration of LPS were established, which were CON group (DMEM), LLPS group (DMEM + 10 μg/mL LPS), 75 MLPS group (DMEM + 50 μg/mL LPS), HLPS group (DMEM + 100 μg/mL LPS),” Your graphs shows up to 200 micro gram/ml of LPS was used. Please clarify.
Response – Thank you for the suggestion. We have made revision. Actually, we only tested the cell viability with the content of 10, 20, 30, 40, 50, 100, 200 μg/mL LPS stimulation. And we chose the content of 0, 10, 50, 100 μg/mL LPS stimulation for further cytokines secretion and gene expression analysis. We did not describe this clearly in the materials and methods, and we have made revision on this part.
“Experiment 1: Lipopolysaccharide (LPS, No: L6143, Salmonella enterica serovar typhimurium) was used to establish an immune-stressed IPI-2I cell (porcine ileum epithelial cells) model. LPS (10, 20, 30, 40, 50, 100, 200μg/mL) was added to the cell culture system for 3 h, 6 h, 9 h or 12 h, respectively. Cell viability was measured using a Cell Counting Kit (CCK-8, No.C0038, Beyotime Btotech Inc, China) according to the manufacturer’s instructions.
Experiment 2: Based on the results of the Experiment 1, four groups with different concentrations of LPS were established: the CON group (DMEM), LLPS group (DMEM + 10 μg/mL LPS), MLPS group (DMEM + 50 μg/mL LPS), and HLPS group (DMEM + 100 μg/mL LPS), with stimulation time of 3 h, 6 h, 9 h, and 12 h. The cell supernatants were collected at different time points for all groups to study cytokine secretion. Additional cells were collected at different time points to study the gene expression of cytokines.”
Comments – 16. viability, Define "viability".
Response – Thank you for the suggestion. “Cell viability was measured using a Cell Counting Kit (CCK-8, No.C0038, Beyotime Btotech Inc, China) according to the manufacturer’s instructions.” We have defined this in the manuscript.
Comments –17. 6h, How did you come up with the doses for LPS (50 micro-gram/ml) and 6 h incubation time? How did you know these were the best doses and times for the treatment?
Response – Thank you for the suggestion. The doses for LPS (50 micro-gram/ml) and 6 h incubation time were based on the results of the Experiment 1 and Experiment 2.
“Experiment 1: Lipopolysaccharide (LPS, No: L6143, Salmonella enterica serovar typhimurium) was used to establish an immune-stressed IPI-2I cell (porcine ileum epithelial cells) model. LPS (10, 20, 30, 40, 50, 100, 200μg/mL) was added to the cell culture system for 3 h, 6 h, 9 h or 12 h, respectively. Cell viability was measured using a Cell Counting Kit (CCK-8, No.C0038, Beyotime Btotech Inc, China) according to the manufacturer’s instructions.
Experiment 2: Based on the results of the Experiment 1, four groups with different concentrations of LPS were established: the CON group (DMEM), LLPS group (DMEM + 10 μg/mL LPS), MLPS group (DMEM + 50 μg/mL LPS), and HLPS group (DMEM + 100 μg/mL LPS), with stimulation time of 3 h, 6 h, 9 h, and 12 h. The cell supernatants were collected at different time points for all groups to study cytokine secretion. Additional cells were collected at different time points to study the gene expression of cytokines.
The results of Experiment 1 and Experiment 2 showed that:
“LPS stimulation (content: ≥ 30 μg/ml; time: 3 h, 6 h or 9 h ) decreased the viability of IPI-2I cells. When the stimulation time was 12 h, LPS at concentrations of 100 μg/ml and 200 μg/ml significantly reduced the viability of IPI-2I cells.
In this study, stimulation with 50 μg/ml LPS for 6 h improved the secretion of IL-1β, TNF-α and IFN-γ and decreased the secretion of IL-10, so as to the genes expression of these cytokines. Hence, 50 μg/ml LPS stimulation for 6 h is suitable for establishing an the immune-stressed model in IPI-2I cells.” We have made revision in the materials and methods to make it more clear.
As for the stimulation time and dose of TTO, “Experiment 3: The regulatory effects of TTO on immune-stressed IPI-2I cells were further studied. The composition of TTO was the same as that published in our previous study [6]. Tea tree oil is a plant essential oil that is difficult to dissolve in high-sugar DMEM. The tea tree oil was filtered through a 0.22μm bactericidal filter into a 50mL centrifuge tube and stored at 4℃ for later use. A DMSO solution with a concentration fraction of 0.01-0.1% TTO was used to stimulate the IPI-2I cells for 3 h, 6 h, 9 h and 12 h. Cell viability was tested.
Experiment 4: According to the results of the cell viability assay, 0.01%, 0.03% and 0.05% TTO were used to stimulate the cells for 6 h.” We have added this information in materials and methods part to make it more clear.
Comments –18. GAPDH, Is it sufficient to have only one housekeeping gene?
Response – Thank you for the suggestion. Actually, we chose two housekeeping gene GAPDH and β-actin. The results were similar. Finally, we chose GAPDH as the housekeeping gene to make the data analysis. And we have made revision in the manuscript.
Comments –19. Primer, Did you test the efficiency of the primers? How did you evaluate the specificity of your primers?
Response – Thank you for the suggestion. We observed the amplification curve, dissolution curve when we conducted the RT-PCR experiment. The amplification curve was normal and the dissolution curve was a single peak, which indicated that the primers have good specificity and efficacy. In addition, agar-gel electrophoresis was conducted after ordinary gradient PCR reaction, the amplification products of the primers were single and were consistent with the expected size, which also indicated that the primer has good specificity. We have made revision to make the description in the manuscript more rigorous.
Comments –20. Statistical Package, what was your experimental unit? what were your random and fixed effects?
Response –Thank you for the suggestion. We only used independent t test and one-way ANOVA in this manuscript. Two-way ANOVA or linear programming model were not used in this manuscript, hence the experiment unit, the random and fixed effect were not included in this manuscript.
Comments –21. delete “extremely”
Response – Thank you for the suggestion. We have made revision to make the description more clear. “All the graphs were drawn using Graphpad Prism 7. The P value indicates the difference among the groups; P<0.05 indicates a significant difference, while P<0.01 indicates the difference is extremely significant.”
Comments –22. delete “respectively”
Response –Thank you for the suggestion and we have deleted “respectively”.
Comments –23. “and the gene expression of IL1β and TNF-α in MLPS group were even higher than th at in the HLPS group” This statement is not consistent with the results shown in the graphs.
Response –Thank you for this suggestion. The results description here is right. We used capital(A, B, C, etc) and lower-case letter (a, b, c, etc)in figures to show P<0.01 and P<0.05, respectively. However, we made labeling error. And we have made revision on the figures throughout the manuscript.
Comments –24. “TTO with the concentration of 0-0.05% had a positive effect on the cell viability of I PI-2I cells; when the culture time was 3h, 9h and 12h, 0.04% TTO significantly increased (P<0.05) the cell viability of IPI-2I cells.” Awkward and confusing statement. This isn't very clear. In the M&M section, you mentioned using three doses of TTO. Here, your graphs show doses. Which one is correct? Please clarify.
Response –Both are correct. “TTO with the concentration of 0-0.05% had a positive effect on the cell viability of IPI-2I cells; when the culture time was 3h, 9h and 12h, 0.04% TTO significantly increased (P<0.05) the cell viability of IPI-2I cells.” This is Experiment 3 before we chose the three doses of TTO to conduct Experiment 4. We need to find out whether the TTO has cytotoxicity. Hence, the results here is very important. And we have made revision in the material and methods part to make the description more clear.
Comments –25. Discussion, The discussion is incomplete. The authors do not provide a mechanistic explanation for the current findings. For instance, they do not explain how TTO down-regulates or up-regulates the expression of the genes.
Response – Thank you for the suggestion. We have made revision. “The effects of TTO on the gene expression of cytokines may be due to the action of terpenes (the main functional component of TTO) on the transcription factor NF-κB [21], and we further discuss the gene expression of NF-κB in the following paragraph.”
Comments –26. “The results of this study will help in the usage of TTO in pig industry” you may not conclude this statement based on the current findings. You have done an in vitro study on epithelial cells. You still need to prove that the same or similar results will be generated in an in vivo study. You may require to establish a new dose.
Response –Thank you for the suggestion. We have made revision. “The results of this study will help to elucidate the underlying mechanisms by which TTO regulates intestinal health.”
Comments –27. “Considering the other items,” What does this mean?
Response –Thank you for the suggestion. We have made revision to make this information more clear. “Considering the other factors including the effects of TTO on the LPS-induced cell apoptosis and cytokine secretion in IPI-2I cells (discussed in the following two paragraphs), 0.03% TTO supplementation is recommended for immune-stressed cell models.”
Comments –28. In conclusion, Is this all you have concluded from the current finding? “The Materials and Methods should be described with sufficient details to allow others to replicate and build on the published results. Please note that the publication of your manuscript implicates that you” This is not a conclusion form the current study. Please delete.
Response –Thank you for the suggestion. We have made revision by deleting these irrelevant sentences.
Comments –29. must make all materials, data, computer code, and protocols associated with the publication available to readers. Please disclose at the submission stage any restrictions on the availability of materials or information. New methods and protocols should be described in detail while well-established methods can be briefly described and appropriately cited.
Response –Thank you for the suggestion. All materials, data and protocols of this manuscript are available to readers. And we have made revision on the materials and methods part to make it more clear. If you have any more question, please contact us, we will try our best to answer any question you posed.

Reviewer 2 Report
Comments and Suggestions for Authors
Dear authors,
The content of the article is very well described, in addition to being interesting for pig production.
I think it's just important that there is a paragraph in the discussions relating the next steps for in vivo testing.
For example, this dose used in vitro cannot be repeated in vivo, right? It would be a very low dose to reach the cell.
I even suggest that the term in vitro appears in the title. I also think it's important that
Author Response
Response to Reviewer 2
Comments – 1. The content of the article is very well described, in addition to being interesting for pig production.
I think it's just important that there is a paragraph in the discussions relating the next steps for in vivo testing.
For example, this dose used in vitro cannot be repeated in vivo, right? It would be a very low dose to reach the cell.
I even suggest that the term in vitro appears in the title. I also think it's important that.
Response – Thank you for the suggestion. Actually, we have done the in vivo testing, the result has published in the paper (Dong et al., 2019). And we have cited this reference in the manuscript. The appropriate dose in vivo is only 1/3 compared to the dose in vitro. That is a very low dose. And considering your suggestion, we have added “in vitro” in the title. The new title is “Protective effects of tea tree oil on inflammatory injury of porcine intestinal epithelial cells induced by lipopolysaccharide in vitro”
Reference:
Dong L. Dietary tea tree oil supplementation improves the intestinal mucosal immunity of weanling piglets. Animal Feed Science and Technology. 2019, 255: 114209.

Reviewer 3 Report
Comments and Suggestions for Authors
Overall evaluation
The work presents numerous typing errors, which denote a lack of attention in the preparation of the manuscript. Too many sentences are isolated from the context, and the English form also needs to be revised. The Authors must explain why the protective effect of TTO was tested only in cells treated with 50 µg/mL LPS when in the stress provocation test the LPS concentration reached 100 µg/mL. The article does not convince me and cannot be published at the moment. It requires a major revision before it can be re-submitted.
Major remarks
· Statistical analysis: the statistical model used is not described and it was not verified whether the data are normally distributed. · Figure 1 shows the use of LPS concentrations of 20 – 40 60 and 200 µg/mL not described in the Materials and Methods section. Furthermore, the data for concentrations 10 – 50 and 100 100 µg/mL are not reported (Lines 70 - 76). Why ? · The cell viability values (expressed as a percentage) with 0 LPS concentration in figure 1 D are much lower than the data at 0 LPS in figure 1 A, why this significant difference in cell viability, already at the start? Since cell viability is expressed as a percentage, it should be equal to 100 at time 0. · Figures 2 and 3. In some incubation times the significance of the differences in the concentration of cytokines is indicated with only one type of capital letter (e.g. A), but in order to be able to talk about significant differences at a given level of significance, they must be use at least 2 different types of capital letters, e.g. A and B. · Figure 4 also shows concentrations different from those reported in Materials and Methods. · Figure 3 shows the values of LPS added to the cell layer medium, however in the Results section the acronyms HLPS for 100 and MLPS for 50 are used instead and this does not make it easier to understand the figure. The authors could also report the corresponding acronym in the figure next to the LPS concentration. · Figure 5 is not clear, if figures 5 A-E are the result of PCA, then in Materials and methods it is necessary to report how this analysis was performed.
Minor remarks
· Line 38. Please use italic font for “salmonella typhimurium”, furthermore “salmonella” must be written using capital letter. · The acronyms TTO and LPS were used without first having defined their meanings.· Line 60. Typing error “ilume” instead of “ileum”.
· Line 68. Please do not begin a sentence with “And”.
· Line 72. Please use italic for “Salmonella enterica serotype typhmurium”.
· Figure 6, captions. There is a typing error: “ther secretion”.
· Figure 7. In the second bar of NKkb statistical significance is represented also with a capital letter (A), but there are not other capital letters (B, C). This is quite strange, have Authors forgotten to reports all significant differences for P <0.01 ?
Comments on the Quality of English LanguageOverall evaluation
The work presents numerous typing errors, which denote a lack of attention in the preparation of the manuscript. Too many sentences are isolated from the context, and the English form also needs to be revised. The Authors must explain why the protective effect of TTO was tested only in cells treated with 50 µg/mL LPS when in the stress provocation test the LPS concentration reached 100 µg/mL. The article does not convince me and cannot be published at the moment. It requires a major revision before it can be re-submitted.
Major remarks
· Statistical analysis: the statistical model used is not described and it was not verified whether the data are normally distributed. · Figure 1 shows the use of LPS concentrations of 20 – 40 60 and 200 µg/mL not described in the Materials and Methods section. Furthermore, the data for concentrations 10 – 50 and 100 100 µg/mL are not reported (Lines 70 - 76). Why ? · The cell viability values (expressed as a percentage) with 0 LPS concentration in figure 1 D are much lower than the data at 0 LPS in figure 1 A, why this significant difference in cell viability, already at the start? Since cell viability is expressed as a percentage, it should be equal to 100 at time 0. · Figures 2 and 3. In some incubation times the significance of the differences in the concentration of cytokines is indicated with only one type of capital letter (e.g. A), but in order to be able to talk about significant differences at a given level of significance, they must be use at least 2 different types of capital letters, e.g. A and B. · Figure 4 also shows concentrations different from those reported in Materials and Methods. · Figure 3 shows the values of LPS added to the cell layer medium, however in the Results section the acronyms HLPS for 100 and MLPS for 50 are used instead and this does not make it easier to understand the figure. The authors could also report the corresponding acronym in the figure next to the LPS concentration. · Figure 5 is not clear, if figures 5 A-E are the result of PCA, then in Materials and methods it is necessary to report how this analysis was performed.
Minor remarks
· Line 38. Please use italic font for “salmonella typhimurium”, furthermore “salmonella” must be written using capital letter. · The acronyms TTO and LPS were used without first having defined their meanings.· Line 60. Typing error “ilume” instead of “ileum”.
· Line 68. Please do not begin a sentence with “And”.
· Line 72. Please use italic for “Salmonella enterica serotype typhmurium”.
· Figure 6, captions. There is a typing error: “ther secretion”.
· Figure 7. In the second bar of NKkb statistical significance is represented also with a capital letter (A), but there are not other capital letters (B, C). This is quite strange, have Authors forgotten to reports all significant differences for P <0.01 ?
Author Response
Response to Reviewer 3
Comments – 1. Overall evaluation
The work presents numerous typing errors, which denote a lack of attention in the preparation of the manuscript. Too many sentences are isolated from the context, and the English form also needs to be revised.
Response – Thank you for the suggestion. We have made revision according to your suggestion. The manuscript has been edited for proper English language, grammar, punctuation, spelling, and overall style by one or more of the highly qualified native English speaking editors at American Journal Experts. The verification code is 37C6-D59E-A9D0-DEEE-5B44.
Comments – 2. The Authors must explain why the protective effect of TTO was tested only in cells treated with 50 µg/mL LPS when in the stress provocation test the LPS concentration reached 100 µg/mL.
Response –Thank you for the suggestion. The doses for LPS (50 µg/ml) and 6 h incubation time were based on the results of the Experiment 1 and Experiment 2.
“Experiment 1: Lipopolysaccharide (LPS, No: L6143, Salmonella enterica serovar typhimurium) was used to establish an immune-stressed IPI-2I cell (porcine ileum epithelial cells) model. LPS (10, 20, 30, 40, 50, 100, 200μg/mL) was added to the cell culture system for 3 h, 6 h, 9 h or 12 h, respectively. Cell viability was measured using a Cell Counting Kit (CCK-8, No.C0038, Beyotime Btotech Inc, China) according to the manufacturer’s instructions.
Experiment 2: Based on the results of the Experiment 1, four groups with different concentrations of LPS were established: the CON group (DMEM), LLPS group (DMEM + 10 μg/mL LPS), MLPS group (DMEM + 50 μg/mL LPS), and HLPS group (DMEM + 100 μg/mL LPS), with stimulation time of 3 h, 6 h, 9 h, and 12 h. The cell supernatants were collected at different time points for all groups to study cytokine secretion. Additional cells were collected at different time points to study the gene expression of cytokines.
The results of Experiment 1 and Experiment 2 showed that:
“LPS stimulation (content: ≥ 30 μg/ml; time: 3 h, 6 h or 9 h ) decreased the viability of IPI-2I cells. When the stimulation time was 12 h, LPS at concentrations of 100 μg/ml and 200 μg/ml significantly reduced the viability of IPI-2I cells.
In this study, stimulation with 50 μg/ml LPS for 6 h improved the secretion of IL-1β, TNF-α and IFN-γ and decreased the secretion of IL-10, so as to the genes expression of these cytokines. Hence, 50 μg/ml LPS stimulation for 6 h is suitable for establishing an the immune-stressed model in IPI-2I cells.” We have made revision in the materials and methods to make it more clear.
Comments – 3.The article does not convince me and cannot be published at the moment. It requires a major revision before it can be re-submitted.Abstract:You need a statement briefly describing your methodology. For example you can say "a gene expression study was conducted to evaluate....."
Response – Thank you for the suggestion. We have made revision by adding a statement briefly describing the methodology in the abstract. “Ileum epithelial cells (IPI-2I) were chosen and an inflammatory injury cell model was generated. The cell viability, cytokine secretion and gene expression of TLR4 and NF-κB were measured to further evaluate the effects of TTO on the inflammatory injury in immune-stressed cells.”
Comments –4. Major remarks
Statistical analysis: the statistical model used is not described and it was not verified whether the data are normally distributed. ·
Response – Thank you for the suggestion. Homogeneity analysis of variance and the normality test were first conducted before Independent t test or Ducan analysis to make sure the data are normally distributed. We have added these information in statistical analysis.
Comments –5. Figure 1 shows the use of LPS concentrations of 20 – 40 60 and 200 µg/mL not described in the Materials and Methods section. Furthermore, the data for concentrations 10 – 50 and 100 100 µg/mL are not reported (Lines 70 - 76). Why ? ·
Response – Thank you for the suggestion. We did not describe the experiment clearly here. Revisions have been made in Materials and Methods section. The revisions are as bellow:
“Experiment 1: Lipopolysaccharide (LPS, No: L6143, Salmonella enterica serovar typhimurium) was used to establish an immune-stressed IPI-2I cell (porcine ileum epithelial cells) model. LPS (10, 20, 30, 40, 50, 100, 200μg/mL) was added to the cell culture system for 3 h, 6 h, 9 h or 12 h, respectively. Cell viability was measured using a Cell Counting Kit (CCK-8, No.C0038, Beyotime Btotech Inc, China) according to the manufacturer’s instructions.
Experiment 2: Based on the results of the Experiment 1, four groups with different concentrations of LPS were established: the CON group (DMEM), LLPS group (DMEM + 10 μg/mL LPS), MLPS group (DMEM + 50 μg/mL LPS), and HLPS group (DMEM + 100 μg/mL LPS), with stimulation time of 3 h, 6 h, 9 h, and 12 h. The cell supernatants were collected at different time points for all groups to study cytokine secretion. Additional cells were collected at different time points to study the gene expression of cytokines.”
Comments –6. The cell viability values (expressed as a percentage) with 0 LPS concentration in figure 1 D are much lower than the data at 0 LPS in figure 1 A, why this significant difference in cell viability, already at the start? Since cell viability is expressed as a percentage, it should be equal to 100 at time 0.
Response –Thank you for the suggestion. We have made revision according to your suggestion to make the cell viability be equal to 100 at time 0.
Comments –7. Figures 2 and 3. In some incubation times the significance of the differences in the concentration of cytokines is indicated with only one type of capital letter (e.g. A), but in order to be able to talk about significant differences at a given level of significance, they must be use at least 2 different types of capital letters, e.g. A and B.
Response –Thank you for the suggestion. We made mistakes here. We have rechecked the data and made revision.
Comments –8. Figure 4 also shows concentrations different from those reported in Materials and Methods.
Response –Thank you for the suggestion. We have made revision in Materials and Methods. “Experiment 3: The regulatory effects of TTO on immune-stressed IPI-2I cells were further studied. The composition of TTO was the same as that published in our previous study [6]. Tea tree oil is a plant essential oil that is difficult to dissolve in high-sugar DMEM. The tea tree oil was filtered through a 0.22μm bactericidal filter into a 50mL centrifuge tube and stored at 4℃ for later use. A DMSO solution with a concentration fraction of 0.01-0.1% TTO was used to stimulate the IPI-2I cells for 3 h, 6 h, 9 h and 12 h. Cell viability was tested.
Experiment 4: According to the results of the cell viability assay, 0.01%, 0.03% and 0.05% TTO were used to stimulate the cells for 6 h. Five groups were established: the CON group (DMEM), LPS group (DMEM + 50 μg/mL LPS), LTTO group (DMEM + 50 μg/mL LPS + 0.01% TTO), MTTO group (DMEM + 50 μg/mL LPS + 0.03% TTO), and the HTTO group (DMEM + 50 μg/mL LPS + 0.05% TTO), with the simultaneous addition of LPS or TTO to the serum-free DMEM, with a treatment time of 6 h. The doses of LPS (50 micro-grams/ml) and the 6 h incubation time were selected based on the results of Experiment 1 and 2, which can be found in the results and discussion section of the manuscript. Cell viability was measured. The cell supernatant and cells were collected for the study of cytokine secretion, cell apoptosis and gene expression of both cytokines and TLR4/NF-κB signaling.”
Comments –9. Figure 3 shows the values of LPS added to the cell layer medium, however in the Results section the acronyms HLPS for 100 and MLPS for 50 are used instead and this does not make it easier to understand the figure. The authors could also report the corresponding acronym in the figure next to the LPS concentration. ·
Response –Thank you for the suggestion. We have made revision in the Figure 3 by adding the corresponding acronym in the figure next to the LPS concentration.
Comments-10. Figure 5 is not clear, if figures 5 A-E are the result of PCA, then in Materials and methods it is necessary to report how this analysis was performed.
Response –Thank you for the suggestion. Figure 5 A-E are representative results of the cell apoptosis tested by flow cytometry. The percentage of apoptotic cells were obtained from the flow cytometry. Three repeats were conducted for each group. The results were analyzed by one-way ANOVA. We have made revision in the Materials and methods.
Comments-11. Minor remarks
- Line 38. Please use italic font for “salmonella typhimurium”, furthermore “salmonella” must be written using capital letter.
Response –Thank you for the suggestion. We have made revision by using “Salmonella typhimurium” to replace “salmonella typhimurium”.
Comments-12. The acronyms TTO and LPS were used without first having defined their meanings.
Response – Thank you for the suggestion. We have made revision by defining the meanings of TTO and LPS when they first appeared.
Comments-13.Line 60. Typing error “ilume” instead of “ileum”.
Response –Thank you for the suggestion. We have made revision by using “ileum” to replace “ilume”.
Comments-14. Line 68. Please do not begin a sentence with “And”.
Response –Thank you for the suggestion. We have deleted the word “And” at the beginning of the sentence.
Comments-15. Line 72. Please use italic for “Salmonella enterica serotype typhmurium”.
Response –Thank you for the suggestion. We have made revision by using italic for “Salmonella enterica serotype typhmurium”.
Comments-16. Figure 6, captions. There is a typing error: “ther secretion”.
Response –Thank you for the suggestion. We have made revision by changing “ther secretion” to “the secretion”.
Comments-17. Figure 7. In the second bar of NKkb statistical significance is represented also with a capital letter (A), but there are not other capital letters (B, C). This is quite strange, have Authors forgotten to reports all significant differences for P <0.01 ?
Response –Yes,we have made a mistake here. And we have made a mistake here. We used capital(A, B, C, etc) and lower-case letter (a, b, c, etc)in figures to show P<0.01 and P<0.05, respectively. However, we made labeling error. And we have made revision on the figures throughout the manuscript.
Comments-18.Comments on the Quality of English Language
Response –The manuscript has been edited for proper English language, grammar, punctuation, spelling, and overall style by one or more of the highly qualified native English speaking editors at American Journal Experts. The verification code is 37C6-D59E-A9D0-DEEE-5B44.

Reviewer 4 Report
Comments and Suggestions for Authors
In this manuscript reported by Li Dong and coworkers investigated the protective effects of TTO on inflammatory injury of porcine intestinal epithelial cells induced by LPS. The study is fitted of the Animals. However, some However, some improvements needed of the current form before publication. Therefore, I recommended the paper for publication after minor revision.
1. The study was aimed to investigate that TOO could alleviate the inflammatory injury of intestinal epithelial cells induced by LPS. Why the authors only choose the ileum epithelial cells?
2. How the tea tree oil dissolve in the cell culture medium?
3. What A, B, C and D are in each figure should be explained under the figure.
4. In the result of cell apoptosis, it is suggested that detailed information of early and late apoptosis should be explained in the section.
5. In line 331-337, the content should be deleted.
6. Whether the LPS and TOO were added to the cells together or separately should be detailed.
Author Response
Response to Reviewer 4
Comments – 1. The study was aimed to investigate that TOO could alleviate the inflammatory injury of intestinal epithelial cells induced by LPS. Why the authors only choose the ileum epithelial cells?
Response –Thank you for the question. When we first established the immune-stressed cell model, we used IPEC-J2 cell lines. However, the viability of IPEC-J2 increased when stimulated by LPS. Considering ileum plays an important part in intestinal mucosal immunity, we tried to use Ileum epithelial cells (IPI-2I) to establish the immune-stressed cell model. The results seems good.
Comments – 2. How the tea tree oil dissolve in the cell culture medium?
Response –Thank you for the suggestion. The tea tree oil was dissolved in the DMSO and then dissolved in the cell culture medium. We have made revision in Materials and Methods section. The revisions are as below:
“Tea tree oil is a kind of plant essential oil, which is difficult to be dissolved in DMEM high-sugar medium. The tea tree oil was filtered through 0.22um bactericidal filter into 50mL centrifuge tube and stored at 4℃ for later use. DMSO solution with concentration fraction 0.01-0.1% TTO was used to stimulate the IPI-2I cells for 3h, 6h, 9h and 12h.”
Comments – 3. What A, B, C and D are in each figure should be explained under the figure.
Response –Thank you for the suggestion. We have added the explanation of A, B, C and D in each figure.
Comments – 4. In the result of cell apoptosis, it is suggested that detailed information of early and late apoptosis should be explained in the section.
Response –Thank you for the suggestion. We have added the detailed information of early and late apoptosis in the results section.
Comments – 5. In line 331-337, the content should be deleted.
Response –Thank you for the suggestion. We have deleted the content in line 331-337.
Comments –6. Whether the LPS and TOO were added to the cells together or separately should be detailed.
Response –Thank you for the suggestion. The LPS and TTO were added to the cells together. We have made revision in the materials and methods to describe this more clearly.

Round 2
Reviewer 3 Report
Comments and Suggestions for Authors
The work has been greatly improved, but in the added sentences there is a typing error error in line 158: the statistical test used is Duncan, not Ducan.
Once this error has been corrected, the manuscript can be accepted.